# Validation of the Children’s Eating Behaviour Questionnaire in Poland

**DOI:** 10.3390/nu14224782

**Published:** 2022-11-11

**Authors:** Żaneta Malczyk, Oliwia Kuczka, Agnieszka Pasztak-Opiłka, Agnieszka Zachurzok

**Affiliations:** 1Chair and Department of Pediatrics, Faculty of Medical Sciences in Zabrze, Medical University of Silesia in Katowice, ul. 3 Maja 13/15, 41-800 Zabrze, Poland; 2Institute of Psychology, Faculty of Social Sciences, University of Silesia, 40-055 Katowice, Poland; 3Navigare Medical Center, 41-300 Dąbrowa Górnicza, Poland

**Keywords:** children, eating behavior, CEBQ, validity, reliability

## Abstract

Introduction: Obesity is increasingly diagnosed in pre-school and early primary school children. Eating styles displayed by the youngest children may contribute to the development of overweight and obesity. Their assessment may be extremely important in diagnosing the causes of obesity, but also in planning treatment. Aim of the study: In view of the need to introduce a tool for assessing eating behaviours in children in Poland, the aim of the study was to develop the Polish adaptation of the Children’s Eating Behaviour Questionnaire (CEBQ). Material and methods: The study group consisted of 151 mothers of children aged 3–10 years (M = 6.77, SD = 2.34), who completed the Polish version of the CEBQ. In order to assess the validity of the questionnaire, a factor analysis was conducted, using the principal components method with the Oblimin rotation and Kaiser normalization. To assess the reliability of the questionnaire, its internal consistency was checked by calculating Cronbach’s alpha consistency coefficient. The external validity of the CEBQ was also checked by correlating its scales with those of the Temperament Questionnaire (EAS-C). Results: The principal components analysis extracted an eight-factor scale from the 35 items of the questionnaire in which a total of 60.57% of the common variance was explained. The validity of such an eight-factor solution was confirmed by the Kaiser method. Satisfactorily high values of Cronbach’s alpha internal consistency coefficient were obtained (0.78). Positive correlations were found between emotionality and emotional undereating and overeating, between shyness and fussiness and negative correlations between activity and slowness in eating, sociability and fussiness and slowness in eating and between shyness and enjoyment of food. Conclusions: The Polish version of the CEBQ is characterized by the acceptable validity and reliability and has a satisfactory criterion accuracy; therefore, it can be used as a psychometric tool to assess eating behaviours in Polish children.

## 1. Introduction

Despite the growing awareness of the topic of nutrition and healthy lifestyle, the number of people struggling with excessive body weight continues to increase [1,2,3]. It is particularly worrying that obesity is increasingly being diagnosed in pre-school and early primary school children [3]. Obesity in childhood is the most challenging public health issue in the twenty-first century. Many specialists consider infancy and adolescence as critical periods for the development of obesity. In most obese infants, the amount of adipose tissue decreases around the age of two, which corresponds with the period when their physical activity increases. In the following years, there is a renewed increase in adipose tissue, called “rebound obesity”, which usually occurs around the age of eight [3].

Numerous studies show that rapid and excessive weight gain during childhood increases the risk of obesity and its complications in adulthood [4]. The prevalence of obesity at the age of six increases the risk of becoming an obese adult by 25% and by 75% at 12 years of age [5].

Even the youngest children exhibit certain eating behaviours, i.e., those that are directly related to meeting nutritional needs. Initially, children form their nutritional behaviour through imitation [6]. This is why the family environment plays such an important role in shaping these behaviours in young children. This is mainly carried out through the daily reinforcement of patterns and specific habits and customs [7]. Kindergartens and schools also have a significant impact on shaping the behaviour related with food consumption. Thoughtful education implies healthy eating knowledge, appropriate attitudes and, consequently, specific lifestyles [8,9,10,11]. The issue of responsible formation of eating behaviours in early childhood is extremely important, as studies show that proper habits continue into adulthood [12]. However, it is suggested that eating behaviours are shaped, not only by environmental factors, but they can be related to temperamental traits. Ashcroft et al. [13] hypothesized that the continuity in the eating styles can be comparable to the continuity observed in longitudinal studies of temperamental traits, such as shyness, activity and emotionality. It was shown that more feeding difficulties are presented in children who are unsociable, difficult or demanding. Moreover, shyness and emotionality can be related to an unwillingness to try a new kinds of food [14].

The differences in eating styles have been hypothesized to be related to both—underweight and overweight. Failure to thrive was shown to be associated with fussy eating. A fussy child may eat too little but also less healthily because it can be problematic to persuade him to eat a healthy diet [15]. Meanwhile, obese children tend to eat faster, are more sensitive to external food signals and have a lower responsiveness to internal satiety signals [16]. So, as eating behaviour patterns become established in the preschool years and tend to remain stable through childhood to the adulthood, excessive body weight may persist from early childhood into the adult years [17].

The most commonly used tool for assessing the nutritional behaviour in children is the CEBQ—Child Eating Behaviour Questionnaire [13,15,16]. It was developed by Wardle et al. in 2001, as a useful tool to measure the eating styles in children, as the early precursors of obesity or eating disorders [15]. The CEBQ has been successfully adapted and used in many countries to measure associations between eating behaviours and body weight in children [17,18,19,20,21,22]. The questionnaire contains 35 questions addressed to parents, covering eight dimensions of eating styles: (1) Food Responsiveness—to measure eating in response to external food cues, (2) Emotional Overeating and (3) Emotional Undereating—to show the increase and decrease eating response to negative emotions (anger, anxiety), (4) Enjoyment of Food—to show the general interest in food, (5) Desire to Drink—to detect the increased desire to drink, especially sweetened drinks, (6) Satiety Responsiveness—to reflect the ability to regulate the amount of eaten food by internal satiety clues, (7) Slowness in Eating—to show the gradually reduced interest in a meal during its eating, and (8) Fussiness—to reflect the unwillingness to try new kinds of food and a lack of interest in food [13,15]. These subscales are usually categorised in food approach (subscales: 1, 2, 4, 5) and food avoidant (subscales: 3, 6, 7, 8).

In the Polish literature, there is a scarce number of studies analysing the impact of eating styles already manifested by the youngest children, on the incidence of obesity [23], moreover there is no tool validated for the Polish population for conducting such studies. The aim of the present study was to develop the Polish adaptation of the CEBQ.

## 2. Materials and Methods

The study was divided into two phases. In the first phase, the original language version of the questionnaire was translated. The translation procedure involved the translation of the content from English into Polish by two independent translators, followed by the creation of one agreed version of the questionnaire in Polish. Subsequently, the Polish version was again translated into English by two bilingual people, using the translation-back translation method. In order to assess the validity of the questionnaire, a factor analysis was conducted using the principal axis method with the Oblimin rotation and Kaiser normalization. To assess the reliability of the questionnaire, its internal consistency was checked by calculating Cronbach’s alpha consistency coefficient. At this stage of the study, the study group consisted of 151 mothers of children aged three–10 years (M = 6.75, SD = 2.34), who completed the CEBQ using the paper-and-pencil method or online. They rated their children’s eating behaviour on a five-point Likert scale (1 = never; 2 = rarely; 3 = sometimes; 4 = often; 5 = always). In five statements, there was an inverse score. The mothers also completed a questionnaire in which they declared their own and their child’s age, weight and height. Based on the information obtained, the BMI was calculated for all (Table 1). The underweight was present in 18 children (12%), overweight in 14 (9%), and the obesity was found in 30 children (20%). In 89 children (59%), the BMI remained within a normal range. The method used in the further part of the work was an exploratory factor analysis. The calculations were performed using the IBM SPSS package.

In the second phase of the study, the external validity of the CEBQ was checked by correlating its scales with the scales of the EAS-C questionnaire (Emotionality, Activity, and Shyness for Children questionnaire) for assessing temperament [24]. The study group consisted of 100 mothers of children aged three to 10 years (M = 6.45, SD = 2.9). The procedure of the study involved the completion of the CEBQ adapted to Polish conditions and the EAS-C temperament questionnaire in the version created for completion by the mothers, covering four temperamental traits: emotionality, activity, sociability and shyness. The data collection took place via the Internet.

## 3. Results

### 3.1. Phase I

In order to check the basis for the factor analysis and the accuracy of the sample selection, the Kaiser–Mayer–Olkin test was used. A KMO > 0.5 indicates the validity of using the exploratory factor analysis [25]. We obtained a satisfactorily high value (KMO = 0.782). In the Bartlett test, the null hypothesis assumes that the correlation matrix is an unitary matrix. If the *p*-value < 0.05, the null hypothesis must be rejected [26]. We gained a statistically significant result (χ2 = 3189.967; *p* < 0.0001), which indicates that the correlation matrix was not a unitary matrix. It also confirms the validity of the use of the factor analysis.

The exploratory factor analysis was carried out using the principal axis method. Due to the assumption of the non-orthogonality of factors, the Oblimin rotation (delta = 0) was used. An eight-factor structure was obtained for the 35 test items. The validity of such an eight-factor solution was confirmed both by the Kaiser method, in which eight factors were given an eigenvalue higher than 1, and by a scree plot.

The eight factors extracted, explained 60.57% of the common variance. The percentage of the explained variance by the individual factors is shown in Table 2.

Table 3 presents the CEBQ scales obtained in the factor analysis, together with the individual items and their factor loadings. The categorisation of a particular item into a factor was accomplished on the basis of an analysis of the factor loadings of the item. It was assumed that a given item is included in a factor if its factor loading value is at least 0.3. In the few cases where the accepted limit was not reached, the item was assigned on the basis of content, based on an analysis of the scales’ reliability. The food responsiveness scale included five items, emotional overeating—four items, emotional undereating—four items, enjoyment of food—four items, desire to drink—three items, satiety responsiveness-four items, slowness in eating—four items, and fussiness—six items. The model turned out to be a good fit for the data. To assess the reliability, the internal consistency was checked by calculating Cronbach’s alpha consistency coefficient. The high reliability of the scale is indicated by the values of this coefficient greater than 0.7. Satisfactory results were obtained for most of the scales, as shown in Table 4, and it ranged from 0.53 for slowness in eating to 0.87 for desire to drink. The overall coefficient of the questionnaire was high and equal to 0.78.

### 3.2. Phase II

Due to the lack of Polish tools assessing eating styles, the criterion relevance was checked by means of a correlation analysis of the EAS-C questionnaire scales, measuring the temperament type with the CEBQ scales [24]. The descriptive characteristics of the distribution of the variables are presented in Table 5.

In order to verify whether there is a correlation between eating styles and temperament traits, a correlation matrix of variables was created, including the CEBQ scales and the EAS-C questionnaire scales. As all variables did not achieve a distribution close to normal, it was decided to use Spearman’s non-parametric rank order correlation. The results are presented in Table 6. The results showed a weak positive correlation between emotionality and emotional undereating and emotional overeating. We did not find any further relationship between emotionality and the CEBQ subscales. A positive correlation was also observed between fussiness and shyness and a negative correlation between fussiness and sociability. A negative correlation was also observed between activity and slowness in eating and slowness in eating and sociability, and a negative correlation between enjoyment of food and shyness.

We also verified the correlation of the multi-choral correlation matrix of the CEBQ scales with the scales of the EAS-C questionnaire. It was decided to use the chi-squared test for the two-dimensional normality. There were no statistically significant correlations between eating styles and temperament traits (*p* > 0.05).

## 4. Discussion

In the Polish literature, there is a scarce number of studies analysing the nutritional behaviours in children, in terms of the prevalence of obesity, mainly due to the lack of a validated tool for conducting such studies. The aim of the present study was to perform a Polish adaptation of the CEBQ. In the first stage of the study, the validity and reliability of the questionnaire were assessed, followed by an evaluation of its external validity. The results obtained indicate that the Polish version of the CEBQ is a useful tool for studying the eating styles in children and has satisfactory psychometric properties. The analysis of the main components identified eight scales, which is in line with the original version of the questionnaire, and the obtained scales had high values of the coefficient of the internal consistency of Cronbach’s alpha coefficient (0.78), which indicates its satisfactory relevance and reliability. There was a slightly lower reliability of the scales in this version than in the original, especially in the satiety responsiveness scale (0.53), which may have its origin in the reported difficulties in understanding some of the scale items, which, however, were included in the final version of the tool for the sake of the content relevance analysis.

To assess the criterion-relevance analysis, the EAS-C temperament questionnaire by Buss and Plomin was used. This questionnaire is based on the genetic theory of temperament, which assumes that temperament is a set of inherited traits that emerge in early childhood [24]. In the version of the questionnaire used, which assumes the description of children’s behaviour by their mothers, four temperamental traits were distinguished: emotionality, activity, sociability and shyness. The decision to use the EAS-C in the criterion-relevance analysis was due to the lack of tools on the Polish market examining constructs similar to eating styles. This illustrates the need for a Polish version of the questionnaire and explains the sense of the work carried out in this direction.

The continuity of the temperamental characteristics from early childhood to adulthood is comparable to the persistence of eating styles in longitudinal studies. Moreover, it was hypothesized that the differences in individuals’ temperament traits may determine why some children are at risk of feeding problems and underweight whereas others have excessive body weight [14]. Furthermore, in our study, we observed the relationship between eating behaviours and temperamental characteristics, suggesting an influence of personal characteristics on eating styles.

The demonstrated correlations between emotionality and emotional undereating and overeating are consistent with previous literature reports [14]. This relationship implies that as the temperament trait emotionality, associated with responding to experienced emotions with anxiety, fear or anger, increases, the tendency to undereat meals or to eat more than usual, increases. This association may be due to personal tendencies to cope with strong emotions and stress through reduced or excessive food intake [27]. However, Blissett et al. suggested that emotional overeating can be associated with the parental use of food to regulate a child’s emotions [28]. The Haycraft’s team conducted a study on a group of 241 mothers of children aged three to eight years. The results indicated that there was a strong relationship between emotionality and eating styles, especially food avoidant behaviours [14]. Moreover, Hafstad et al., while conducting a study on a group of children under observation from two to five years of age, proved that a difficult temperament at a young age was a predictor of later pickiness and selectivity in eating [29]. In contrast to other authors [14], we did not find any other relationships between emotionality and the CEBQ subscales, what suggests that a child’s emotionality is not related to every eating problem but mainly to emotional food approach. In emotional children, the relationship with emotional overeating and undereating can indicate eating problems and undesirable weight changes. Haycraft et al. showed that emotional overeating is related positively with a child’s BMI [14].

According to the literature, the rate of eating is influenced by environmental, biological and genetic factors, as well as temperament, which may explain the negative correlation found in our study between activity and slowness in eating [30]. A negative correlation was also found between slowness in eating and sociability. These correlations indicate that the higher the activity, related to various motor activities of the child, the lower the tendency to eat slowly, and the higher the sociability, the higher the tendency to eat quickly. This may be due to the fact that children who enjoy being in the company of others, playing actively, may eat more quickly so as not to limit play time with their peers.

Research findings on the influence of sociability on fussiness are divergent. There are reports showing them to be mutually correlated [15,16,31], as well as some in which these relationships have not been observed [14]. The negative correlation found in the present study between sociability and fussiness certainly requires further research. It can be speculated that children who are more open to others (more sociable), will also be more willing to try new foods and accept new tastes. Moreover, the demonstrated negative correlation between shyness and enjoyment of food is consistent with literature data. Pilner et al. showed an association between shyness and aversion to eating and trying new foods [31]. It means that as shyness, which is afear and aversion to new people, increases, the enjoyment and interest associated with meals, which are included in the enjoyment of food scale, decreases. Additionally, in our study, we found a significant relationship, so far undescribed, between fussiness and shyness. It seems to be consistent with the previously described negative relationship between enjoyment of food and shyness. It may indicate that shy children often show an aversion to novelty, are more likely to refuse food and are more selective about it, which may result in greater fussiness. These relationships, between eating behaviours and sociability, shyness and activity, showed that social factors can significantly influence the eating styles of the young children.

The limitations of the current study should be mentioned. The CEBQ only provides a maternal, subjective perception of the child’s eating behaviour and could be a subject of bias. We still lack objective measures of the other eating behaviours. Moreover, the study used maternal reporting to obtain height and weight of children and mothers, which may be related to the over- or underestimation of relevant indicators. We decided not to use this indicator in the correlations assessment between eating behaviours and children’s BMI.

## 5. Conclusions

The Polish version of the CEBQ has an acceptable validity and reliability and has a satisfactory criterion validity, and can therefore be used as a psychometric tool to assess eating behaviours in children.

The demonstrated correlations between temperament and eating styles show that an early identification of behaviours related to inherited temperamental traits that emerge in early childhood, may be helpful in the identification of children at risk of disturbed eating behaviours and undesirable weight changes.

## Figures and Tables

**Table 1 nutrients-14-04782-t001:** Descriptive statistics for the variables: mother’s age, child’s age, mother’s BMI, child’s BMI centile.

	Min	Max	Mean	SD
Mother’s age [years]	21.0	49.0	34.6	5.6
Child’s age [years]	3.0	10.0	6.75	2.3
Mother’s BMI [kg/m^2^]	16.9	45.63	24.5	4.3
Child’s BMI centile	0.1	99.9	53.8	34.0

SD—standard deviation.

**Table 2 nutrients-14-04782-t002:** Variance explained by the individual factors of the CEBQ.

Factor	% Explained Variance
Food Responsiveness	20.53
Emotional Overeating	4.48
Emotional Undereating	12.57
Enjoyment of Food	1.85
Desire to Drink	5.78
Satiety Responsiveness	3.78
Slowness in Eating	2.70
Fussiness	8.97

**Table 3 nutrients-14-04782-t003:** Factor loadings of the CEBQ items.

Factor	Item	Factor Loading
Food Responsiveness	12—My child always asks for food	0.47
	14—If allowed to, my child would eat too much	0.62
	19—Given the choice, my child would eat most of the time	0.72
	28—Even if my child is full, he/she finds room to eat his/her favourite food	0.51
	34—If given the chance, my child would always have food in his/her mouth	0.80
Emotional Overeating	2—My child eats more when he/she is worried	−0.73
	13—My child eats more when he/she is annoyed	−0.85
	15—My child eats more when he/she is anxious	−0.87
	27—My child eats more when he/she has nothing else to do	−0.32
Emotional Undereating	9—My child eats less when she/he is angry	0.77
	11—My child eats less when she/he is tired	0.68
	23—My child eats more when she/he is happy	0.66
	25—My child eats less when he/she is upset	0.78
Enjoyment of Food	1—My child loves food	0.88
	5—My child is interested in food	0.72
	20—My child looks forward to mealtimes	0.5
	22—My child enjoys eating	0.74
Desire to Drink	6—My child is always asking for a drink	−0.73
	29—If given the chance, my child would drink continuously throughout the day	−0.8
	31—If given the chance, my child would always have a drink	−0.97
Satiety Responsiveness	3—My child has a big appetite	0.52
	17—My child leaves food on his/her plate at the end of a meal	−0.24
	21—My child gets full before his/her meal is finished	−0.28
	26—My child gets full easily	−0.4
	30—My child cannot eat a meal if he/she has had a snack just before	−0.25
Slowness in Eating	4—My child finishes his/her meal very quickly	−0.23
	8—My child eats slowly	0.64
	18—My child takes more than 30 min to finish a meal	0.66
	35—My child eats more and more slowly during the course of a meal	0.5
Fussiness	7—My child initially refuses to try new foods	−0.7
	10—My child eats less when angry	0.79
	16—My child enjoys a wide variety of foods	0.74
	24—My child is difficult to please with a meal	−0.39
	32—My child is willing to try unfamiliar products	0.8
	33—My child says he/she does not like certain products, although he/she has not tried them before	−0.44

**Table 4 nutrients-14-04782-t004:** Cronbach’s α internal consistency of the CEBQ.

Scale	Mean (SD)	Cronbach’s α
Food Responsiveness	2.47 (0.93)	0.84
Emotional Overeating	2.09 (0.84)	0.83
Emotional Undereating	2.65 (0.93)	0.81
Enjoyment of Food	3.3 (0.89)	0.85
Desire to Drink	2.76 (0.96)	0.87
Satiety Responsiveness	2.93 (0.65)	0.53
Slowness in Eating	2.95 (0.80)	0.64
Fussiness	3.05 (0.96)	0.85

SD—standard deviation.

**Table 5 nutrients-14-04782-t005:** Descriptive statistics for the quantitative variables in stage II of the study.

Variable	Min	Max	Mean	Median	SD	Skewness	Kurtosis	K-S d	*p*
FR	5	25	12.02	11	4.63	0.87	0.68	0.13	0.1
EO	3	15	6.72	6	2.45	0.83	0.62	0.17 **	0.01
EN	4	20	10.75	10	3.95	0.57	0.14	0.14	0.1
EW	4	20	12.41	12	3.86	−0.001	−0.56	0.1	0.2
DD	3	15	8.90	8	2.82	0.45	−0.43	0.15 *	0.05
RS	5	25	14.78	15	3.83	0.18	0.26	0.11	0.2
SE	5	20	11.43	11	3.33	0.41	−0.36	0.11	0.2
F	6	30	18.96	20	6.00	−0.23	−0.54	0.08	0.2
E	5	25	15.04	15	5.24	−0.01	−0.71	0.06	0.2
A	5	25	19.1	20	4.86	−0.76	−0.23	0.15 *	0.05
So	5	25	17.76	18	4.66	−0.53	−0.77	0.12	0.15
Sh	5	22	12.1	11.5	4.34	0.25	−0.77	0.1	0.2

FR—food responsiveness; EO—emotional overeating; EN—emotional undereating; EW—enjoyment of food; DD—desire to drink; RS—satiety responsiveness, SE—slowness in eating; F—fussiness; E—emotionality; A—activity; So—sociality; Sh—shyness; SD—standard deviation; *p*—level of significance; K-S—Kolomogorov–Smirnov test. * *p* < 0.05; ** *p* < 0.01.

**Table 6 nutrients-14-04782-t006:** Correlation matrix of the CEBQ scales with the EAS-C questionnaire scales.

Variable	Emotionality	Activity	Sociability	Shyness
Food Responsiveness	0.09	0.06	0.17	−0.08
Emotional Overeating	0.31 **	0.11	0.18	−0.06
Emotional Undereating	0.24 *	−0.12	−0.09	0.05
Enjoyment of Food	0.07	0.08	0.19	−0.20 *
Desire to Drink	0.10	−0.003	0.03	−0.05
Satiety Responsiveness	0.03	−0.17	−0.15	0.05
Slowness in Eating	−0.12	−0.34 **	−0.31 **	0.18
Fussiness	−0.08	−0.03	−0.33 **	0.23 *

* *p* < 0.05; ** *p* < 0.01

## Data Availability

The raw data supporting the conclusions of this article will be made available by the authors without undue reservation.

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
