# Peer review of "Validation of the Children’s Eating Behaviour Questionnaire in Poland"

_nutrients, 2022, doi:10.3390/nu14224782_

Round 1

Reviewer 1 Report

Materials and Methods: You need to include a description of the study group. You mention them briefly in the abstract (line 17-19), but this information also belongs in the paper itself. Overall, very nice paper

Author Response

Dear Reviewer,

Thank You for your valuable comments and feedback provided to our article which will allow to present even better quality paper.

Based on your comments we have included a description of the study group in material and methods section (red font). In the first phase the study group consisted of 151 mothers of children aged 3-10 years (M=6.75, SD=2.34), who completed the CEBQ by paper-and-pencil method or online. In the second phase of the study the study group consisted of 100 mothers of children aged 3 to 10 years (M=6.45, SD=2.9).

Reviewer 2 Report

The authors show the validation of CEBQ in Poland.

Please check:

The value of the index as Cronbach of alpha should also be reported in the abstract

Lines 62-72 since you explain and describe the original questionnaire you have to go to the materials section.

There are no tables. Describe the sample used to obtain data.

In Phase I please report the bibliography of the indexes (KMO and Bartlett) and try to explain after reviewing the whole process

In Phase II, check whether there is a possibility to implement also the polychoric correlation and what results could be given.

Please review these two paragraphs and describe only the methodology used to place the final results in a section of the appropriate results. Please report the limits and put them in the discussion section

Author Response

Dear Reviewer,

Thank You for your valuable comments and feedback provided to our article which will allow to present even better quality paper. We as authors, carefully considered the comments and responded to each of them. Here are the point-by-point answers:

- we reported the value of the index Cronbach of alpha in the abstract (red font)

- we described the translation procedure of the original questionnaire in material and methods section

- we described the sample used to obtain data in material and methods section

- we have included tables

- in Phase I we added the bibliography of the indexes (KMO and Bartlett) and we explained the statistic process

- in Phase II we verified the correlation of the multi-choral correlation matrix of CEBQ scales with the scales of the EAS-C questionnaire. There were no statistically significant correlations between eating style and temperament traits (p> 0.05). The results are presented in Table below:

Variable

Emotionality

Activity 

Sociability

Shyness

Food Responsiveness

0.11 (0.10)

0.05 (0.10)

0.19 (0.10)

-0.08 (0.10)

Emotional Overeating

0.37 (0.08)

0.12 (0.10)

0.18 (0.10)

-0.08 (0.10)

Emotional Undereating

0.28 (0.09)

-0.11 (0.10)

-0.07 (0.10)

0.02 (0.10)

Enjoyment of Food

0.05 (0.10)

0.10 (0.10)

0.18 (0.10)

-0.19 (0.10)

Desire to Drinks 

0.07 (0.10)

-0.02 (0.10)

0.01 (0.10)

-0.06 (0.10)

Satiety Responsiveness  

0.02 (0.10)

-0.21 (0.10)

-0.22 (0.10)

0.10 (0.10)

Slowness in Eating

-0.11 (0.10)

-0.38 (0.08)

-0.29 (0.09)

0.19 (0.10)

Fussiness

-0.08 (0.10)

-0.05 (0.10)

-0.38 (0.08)

0.26 (0.09)

p > 0.05

- we have reported the limits and included them in the discussion section

Round 2

Reviewer 2 Report

The authors provided all the requests.

The only thing that will be better to improve is to review the last sentence in line 103 giving more precise details  "Data collection took place via the Internet."